# Two-dimensional numerical simulations of vortex-induced vibrations for a cylinder in conditions representative of wind turbine towers

Axelle Viré[1], Adriaan Derksen[1,2], Mikko Folkersma[1], and Kumayl Sarwar[2]

[1]Wind Energy Section, Faculty of Aerospace Engineering, Delft University of Technology, The Netherlands
[2]Siemens Gamesa Renewable Energy, Den Haag, The Netherlands
**Correspondence:** Axelle Viré (a.c.vire@tudelft.nl)

**Abstract.**

Vortex-induced vibrations (VIV) of wind turbine towers can be critical during the installation phase, when the rotor-nacelle assembly is not yet mounted on the tower. The present work uses numerical simulations to study VIV of a two-dimensional cylinder in the transverse direction, under flow conditions that are representative of wind turbine towers, both from a fluid-dynamics and structural-dynamics perspective. First, the numerical tools and fluid-structure interaction algorithm are validated by considering a cylinder vibrating freely in a laminar flow. In that case, both the motion amplitude and frequency are shown to agree well with previous results from the literature. Second, VIV is modelled in the turbulent supercritical regime using Unsteady Reynolds-Averaged Navier–Stokes equations. In this context, the turbulence model is first validated on flow past a stationary cylinder at high Reynolds number. Then, results from forced vibrations are validated against experimental results for a range of reduced frequencies and velocities. It is shown that the behaviour of the aerodynamic damping changes with the frequency ratio, and can therefore lead to either self-limiting or self-exciting VIV when the cylinder is left to freely vibrate. Finally, results are shown for a freely-vibrating cylinder under realistic flow and structural conditions. While a clear lock-in map is identified and shows good agreement with published numerical and experimental data, the work also highlights the unsteady nature of the aerodynamic forces and motion under certain operating conditions.

## 1 Introduction

Vortex-induced vibrations (VIV) are structural vibrations that can occur due to the shedding of flow vortices when a fluid flow passes around a structure. Due to this fluid-structure interaction (FSI) phenomenon, a synchronization (also called lock-in) of the vortex shedding and the structural motion can occur for certain flow conditions and/or structural properties, leading to premature fatigue failures. Vortex-induced vibrations occur in many engineering applications, such as suspension bridges, marine risers, and industrial chimneys. In the context of wind energy, such vibrations have been observed numerically both on the tower (Livanos, 2018; Derksen, 2019) and the wind turbine rotor (Heinz et al., 2016; Horcas et al., 2019). The increase in rotor height and diameter makes VIV an increasingly problematic issue when designing the new generation of wind turbines. This work is motivated by the specific problem of VIV of wind turbine towers during their installation. In that phase, when the

tower does not yet support the rotor-nacelle assembly, it acts as a beam clamped at one of its ends and is subjected to a wind flow. Because of the vortex shedding developing in the tower wake, the tower may start to oscillate. This is particularly critical at three different stages of the installation process, namely when the tower: (i) stands on the quay-side, (ii) is transported on a vessel offshore, and (iii) is installed on the offshore foundation in the absence of rotor-nacelle assembly.

The design of wind turbine towers depends on a number of parameters (Dykes et al., 2018). The tower natural frequency, which is important for the dynamic response of the structure to vortex-induced excitations, depends on the tower diameter, thickness, material, and geometry (taper). Various combinations of these parameters are therefore critical for the occurrence of VIV. For the current towers used by Siemens Gamesa Renewable Energy, it is observed that tower diameters in the range of $4.5\text{m} \leq D \leq 6.5\text{m}$ and first bending frequency $0.5\text{Hz} \leq \omega_{bm1} \leq 1.1\text{Hz}$ are most critical for the occurrence of VIV. However, as mentioned above, this is thickness-, material-, and shape- dependent. Furthermore, the trend over the years has been to reduce the tower clamped first mode frequency. This brings challenges as the critical resonance velocities are at a lower wind speed, which results in higher VIV incidences.

In practice, wind turbine towers are usually tapered cylinders with sections of discrete diameters. This shape, together with wind shear, might influence the VIV behaviour of the structure compared to that of circular cylinders. This has already been investigated to some extend in the literature (Balasubramanian et al., 2001; Hover et al., 1998; Bourguet et al., 2013; Bourguet and Triantafyllou, 2016). In this paper, however, this effect is neglected and circular cylinders are considered instead, with a focus on large Reynolds numbers. Despite this limitation, the present study is relevant for the wind energy industry because of the interest of the wind turbine developers for reducing the tapering at the top sections. The phenomenological aspects of VIV for circular cylinders have been extensively analysed in the literature since several decades ago. However, studies at large Reynolds numbers and large mass ratios (defined as the ratio between the structure mass and the mass of the displaced fluid) are still quite rare. Experiments in both air (Feng, 1968; Brika and Laneville, 1993) and water (Govardghan and Williamson, 2000) showed that cylinders with large mass ratios exhibit much smaller amplitudes of vibrations that those with low mass ratios. Belloli et al. (2012) however showed that this is not necessarily true under large Reynolds numbers of the order of $Re = 50,000$. In this context, the maximum non-dimensional oscillation amplitude can become larger than the cylinder diameter. This was also described in other studies with low mass ratios (Govardghan and Williamson, 2006). For this reason, the original Griffin plot (Griffin, 1980), describing the maximum oscillation amplitude as a function of the mass damping parameter, needs to be modified for large Reynolds numbers.

It is worth noting that all these VIV studies were limited to the subcritical regime, in which the boundary layer remains fully laminar and the drag coefficient is nearly constant with the Reynolds number. By contrast, wind turbine towers experience flows both in the transitional regime ($1.5 \cdot 10^5 \leq Re \leq 3.5 \cdot 10^6$), where laminar separation bubbles can exist and reduce the drag coefficient, and in the supercritical regime ($Re \geq 3.6 \cdot 10^6$), in which the boundary layer is fully turbulent. Experiments in these regimes can be challenging due to the need for large wind tunnels and wind speeds. In the context of stationary cylinders, a series of experiments have been performed at these very large Reynolds numbers (Roshko, 1961; Achenbach, 1968; Jones et al., 1969; van Nunen, 1974; Schewe, 1983). These studies generally agree well on the value of the separation angle and the location of the minimum pressure coefficient. However, large discrepancies are found between the various experimental

data for the values of the aerodynamic quantities, such as pressure coefficient, lift and drag coefficients, and Strouhal number.

Jones et al. (1969) also looked at a circular cylinder under forced vibration. Their study showed that a lift amplification exists and is due to the cylinder motion, in agreement with other works. Numerical studies have also been performed for flow past 2D and 3D stationary cylinders. These include Unsteady Reynolds Navier–Stokes (URANS) models (Travin et al., 2000; Catalano et al., 2003; Ong et al., 2009; Rosetti et al., 2012), large-eddy simulations (LES) (Breuer, 2000; Catalano et al., 2003; Singh and Mittal, 2005) and detached-eddy simulations (DES) (Travin et al., 2000; Squires et al., 2008). Scatter has

been observed between the various URANS simulations and was attributed to either different wall function implementations or the fact that some simulations were done in 2D instead of 3D. In most cases, the numerical results also deviate from the experiments, especially with URANS. The latter methodology indeed presents shortcomings, such as an isotropic eddy viscosity, homogeneous Reynolds stresses, and the modelling of the full range of turbulent eddies (Rosetti et al., 2012). Yet, it was found that URANS can still lead to satisfactory engineering results in the supercritical flow regime (Ong et al., 2009).

The present study will go a step further by presenting URANS results of a freely-vibrating cylinder in the supercritical regime ($Re \geq 3.6 \cdot 10^6$). To the best of the authors' knowledge, this is the first numerical work to do so.

The paper is organised as follows. Section 2 presents the fluid-dynamics and structural-dynamics models used in this study. It also explains the algorithm used to couple these models and perform two-way coupled fluid-structure interaction (FSI) simulations of a freely-vibrating cylinder. Section 3 shows the results for two different regimes. First, laminar flow is considered

in order to validate both the models and the FSI algorithm. Results of a freely-vibrating cylinder are presented and compared to the literature. Second, the turbulent supercritical regime is considered for three cases: stationary cylinder, cylinder undergoing forced vibrations, and cylinder undergoing free vibrations. Finally, conclusions are drawn in Section 4.

## 2  Computational approach

### 2.1  Fluid dynamics model

The computational fluid dynamics (CFD) model used in this study is the open-source code Open-FOAM-v1812. The model uses a finite-volume discretisation of the incompressible Navier–Stokes equations for a Newtonian fluid. In the laminar regime, these equations are solved directly, without using a turbulence model, through direct numerical simulations (DNS). In the turbulent regime, the Unsteady Reynolds-Averaged Navier–Stokes (URANS) equations are solved using a k-$\omega$ SST turbulence model (Menter, 1994), namely

$$\frac{\partial \overline{\boldsymbol{u}}}{\partial t} + \nabla \cdot (\overline{\boldsymbol{u}\boldsymbol{u}}) = -\frac{\nabla \overline{p}}{\rho} + \nabla \cdot (\nu \nabla \overline{\boldsymbol{u}}) - \nabla \cdot (\overline{\boldsymbol{u}'\boldsymbol{u}'}), \tag{1}$$

$$\nabla \cdot \overline{\boldsymbol{u}} = 0, \tag{2}$$

where the fluid velocity is decomposed into an averaged and a fluctuating part, i.e. $\boldsymbol{u}(\boldsymbol{x},t) = \overline{\boldsymbol{u}}(\boldsymbol{x},t) + \boldsymbol{u}'(\boldsymbol{x},t)$ with $\overline{\boldsymbol{u}'} = 0$. Furthermore, $p$ is the pressure field, $\rho$ is the fluid density, and $\nu$ is the fluid kinematic viscosity. Additionally, the turbulence model assumes that

$$-\overline{u_i' u_j'} = 2\nu_t S_{ij} - \frac{2}{3}\delta_{ij} k, \tag{3}$$

with $\delta_{ij}$ denoting the Kronecker symbol, $\nu_t$ the eddy viscosity, $S_{ij}$ the strain rate tensor defined as

$$S_{ij} = \frac{1}{2}\left(\frac{\partial \overline{u}_i}{\partial x_j} + \frac{\partial \overline{u}_j}{\partial x_i}\right) - \frac{1}{3}\delta_{ij}\frac{\partial \overline{u}_k}{\partial x_k}, \tag{4}$$

and $k$ the turbulent kinetic energy as

$$k = \frac{1}{2}\overline{u_i' u_i'}. \tag{5}$$

The specific implementation of the model is that presented by Menter et al. (2003), with a revised turbulence specific dissipation rate production term from Menter and Esch (2001). The turbulent specific dissipation rate $\omega$ is computed by solving the following equation,

$$\frac{\partial \omega}{\partial t} + \frac{\partial u_j \omega}{\partial x_j} = \gamma \frac{\tilde{P}_k}{\nu_t} - \beta \omega^2 + \frac{\partial}{\partial x_j}\left[(\nu + \sigma_\omega \nu_t)\frac{\partial \omega}{\partial x_j}\right] + 2(1-F_1)\sigma_{\omega_2}\frac{1}{\omega}\frac{\partial k}{\partial x_j}\frac{\partial \omega}{\partial x_j}, \tag{6}$$

where the blending function $F_1$ is given by

$$F_1 = \tanh\left[\left[\min\left[\max\left(\frac{\sqrt{k}}{\beta^* \omega \gamma}, \frac{500\nu}{\gamma^2 \omega}\right), \frac{4\sigma_{\omega_2} k}{CD_{k\omega}\gamma^2}\right]\right]^4\right], \tag{7}$$

and

$$CD_{k\omega} = \max\left(2\sigma_{\omega_2}\frac{1}{\omega}\frac{\partial k}{\partial x_j}\frac{\partial w}{\partial x_j}, 10^{-10}\right). \tag{8}$$

The transport equation for the turbulent kinetic energy $k$ is given by

$$\frac{\partial k}{\partial t} + \frac{\partial u_j k}{\partial x_j} = \tilde{P}_k - \beta^* k\omega + \frac{\partial}{\partial x_j}\left[(\nu + \sigma_k \nu_t)\frac{\partial k}{\partial x_j}\right], \tag{9}$$

where the limited production term $\tilde{P}_k$ is given by

$$\tilde{P}_k = \min(P_k, 10\beta^* k\omega), \tag{10}$$

and

$$P_k = \nu_t \frac{\partial u_i}{\partial x_j}\left(\frac{\partial u_i}{\partial x_j} + \frac{\partial u_j}{\partial x_i}\right). \tag{11}$$

**Table 1.** Turbulence coefficients for the present k-$\omega$ SST model.

| Model coefficient | Value [–] |
|:---:|:---:|
| $\sigma_{k_1}$ | 0.85 |
| $\sigma_{k_2}$ | 1 |
| $\sigma_{\omega_1}$ | 0.5 |
| $\sigma_{\omega_2}$ | 0.856 |
| $\beta_1$ | 3/40 |
| $\beta_2$ | 0.0828 |
| $\gamma_1$ | 5/9 |
| $\gamma_2$ | 0.44 |
| $\beta^*$ | 0.09 |
| $a_1$ | 0.31 |

Furthermore, the values of the turbulence coefficients are given in Tab. 1. If needed, the constants are blended (typically close to the boundary layer) by the following interpolation,

$$\phi = F_1\phi_1 + (1 - F_1)\phi_2, \tag{12}$$

where $\phi$ represents any of the coefficients in Tab. 1. Once the two turbulence transport equations are solved, the eddy viscosity field is obtained as

$$\nu_t = \frac{a_1 k}{\max(a_1\omega, \Omega F_2)}, \tag{13}$$

in which $\Omega$ is the magnitude of the strain rate tensor, $a_1$ is a coefficient defined in Tab. 1, and $F_2$ is the following blending function,

$$F_2 = \tanh\left[\left[\max\left(\frac{2\sqrt{k}}{\beta^*\omega\gamma}, \frac{500\nu}{\gamma^2\omega}\right)\right]^2\right]. \tag{14}$$

The equations are discretised using second-order methods in both space and time. Only the equations for $k$ and $\omega$ are solved using a first-order upwind scheme. The PIMPLE algorithm is used for the pressure-velocity coupling. This model represents the state-of-the-art in URANS modelling. However, it does have the limitation that the model is two-dimensional, and therefore, cannot take vertical wind shear into account. Also, it has some limitations for the boundary layer transition regime.

## 2.2 Structural dynamics model

The two-dimensional cylinder is modelled as a rigid body of mass $m$ attached to a linear spring, with spring constant $k_s$, and a viscous damper, with damping coefficient $c$. The motion of the rigid body is constrained so that it cannot rotate and can only

125 translate in the y-direction transverse to the inflow. The response of the body is therefore governed by the equations of motion of a single degree-of-freedom oscillator, i.e.

$$m\ddot{y} + c\dot{y} + k_s y = F_y,\tag{15}$$

where $\ddot{y}$, $\dot{y}$, and $y$ are the y-component of the linear acceleration, velocity and displacement, respectively, of the center of gravity of the structure and $F_y$ is the resultant traction force exerted on the structure by the fluid. Equation (15) can be written

130 as

$$\ddot{y} + 2\zeta\omega_n\dot{y} + \omega_n^2 y = \frac{F_y}{m},\tag{16}$$

where $\omega_n = \sqrt{k_s/m}$ is the natural frequency of the structure and $\zeta$ is the damping coefficient defined as the ratio between the actual damping and the critical value, i.e.

$$\zeta = \frac{c}{2\sqrt{k_s m}}.\tag{17}$$

135 For a three-dimensional tower, the values for mass, stiffness and damping vary along the tower. Here, a modal analysis was performed to lump these values into a single value representing the two-dimensional property of the cylinder. To do so, the three-dimensional tower is divided into 40 segments with 6 degrees-of-freedom at each node, leading to stiffness and mass matrices of dimensions $246 \times 246$. The natural frequency of the first mode is obtained by solving Eq.(15) in the frequency domain. The modal mass and stiffness are then obtained by pre-multiplying and post-multiplying the mass and

140 stiffness matrices by the eigenvector of the first bending mode. Equation (16) is further non-dimensionalised as

$$\ddot{y}^* + 2\zeta\dot{y}^* + y^* = 2nU^* C_{F_{y^*}},\tag{18}$$

denoting $y^* = y/D$ ($D$ being the cylinder diameter), $t^* = t/\omega_n$, and $m^* = 2m/(\rho D)$. Additionally, $C_{F_{y^*}}$ stands for the aerodynamic coefficient expressed in terms of $y^*$, $n = \rho D^2/(4m)$, and the reduced velocity is $U^* = U/(D\omega_n)$.

## 2.3 Fluid-structure interaction coupling algorithm

145 In this work, fluid-structure interactions are modelled using a partitioned approach, whereby the fluid and structural dynamics are solved alternatively at every time step. Here, both weakly- and strongly- coupled schemes are considered. This is illustrated by Fig. 1. At the start of a new time step, Eq. (15) is solved for the structural dynamics. Once the structure displacement is known, the geometry is moved accordingly and the mesh is diffused using a spherical linear interpolation scheme (SLERP) algorithm. With this approach, it is possible to specify a mesh region where the cells preserve their shape during motion. In

150 this work, for the freely-vibrating cases, the mesh in a distance of up to $25D$ around the cylinder is kept rigid throughout the simulations, in order to keep its initial high-quality characteristics in the O-grid block. By contrast, the mesh deformation is applied to all cells located further away from the cylinder. Once the mesh has moved, the fluid dynamics solver solves the Navier–Stokes equations accounting for the mesh motion using an Arbitrarily Lagrangian Eulerian approach. In the weakly-coupled algorithm, the fluid-dynamics equations are solved iteratively without solving again for the structural dynamics. As

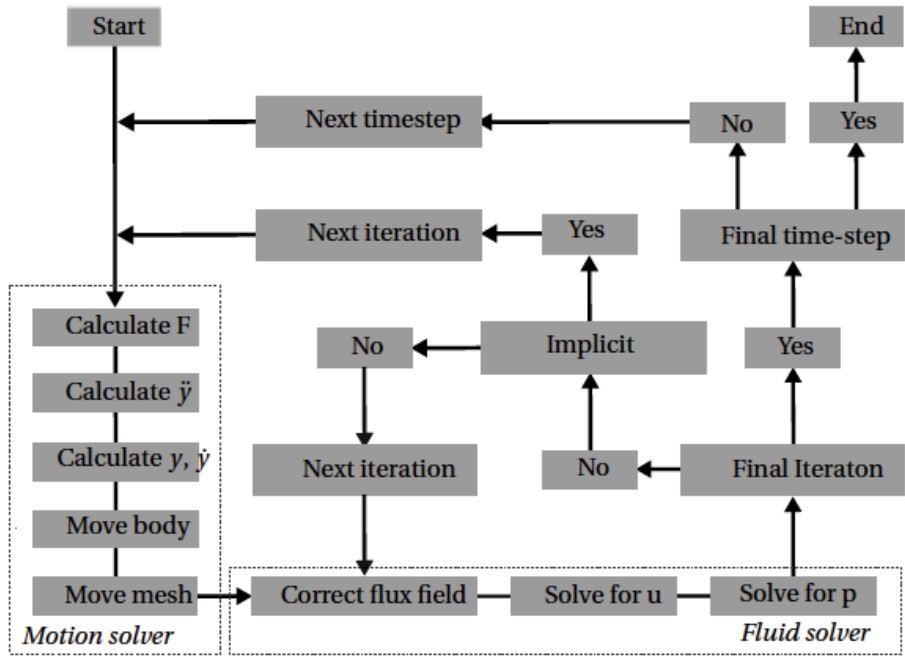

**Figure 1.** Sketch of the coupling weakly- (explicit) and strongly- (implicit) coupled algorithms between structure and fluid solvers. Figure taken from Derksen (2019).

such, the structural and mesh motion are only solved once per time step. By contrast, in the strongly-coupled approach, the structural dynamics equations are solved for each sub-iteration of the fluid solver. Thus, the structural dynamics and mesh movement are solved multiple times per time step.

In this work, a weakly-coupled algorithm was found to give accurate results for modelling VIV in the laminar regime. In that case, the structural dynamics equations were also solved using an explicit solver, the so-called symplectic integrator (Dullweber et al., 1997), which is based on the leapfrog method. The explicit character of this solver leads to a constant structural displacement for each iteration within a time step, while the acceleration and velocity may change at each iteration. The structural displacement of the current time step is only dependent on the acceleration computed at the previous time step. In the turbulent regime, a strongly-coupled FSI scheme was adopted and 3 to 4 sub-iterations were necessary between fluid and structural solvers to achieve accurate results. In that case, an implicit Newmark structural solver was further used with the so-called average constant acceleration method (Newmark, 1959).

## 3 Results

The results are shown for two distinct flow regimes. First, a freely-vibrating cylinder is considered in a laminar flow in order to validate both the setup and the fluid-structure interaction algorithm. Second, the results are presented in a turbulent regime relevant for wind turbines. In that case, stationary cylinder, forced vibrations, and free vibrations are considered.

### 3.1 Laminar flow

#### 3.1.1 Simulation setup

The computational domain used in the present work is sketched in Fig. 2. The cylinder is centered in a squared domain of length $50D$, $D$ being the cylinder diameter. A uniform streamwise velocity field is imposed as initial condition and also as a Dirichlet boundary condition at the inlet of the domain. A zero-pressure Dirichlet condition is set at the outlet. The lateral far-field boundaries are set as slip walls. In this subsection, free-vibration results are shown whereby the dynamics of the cylinder is determined by solving Eq. (15). The domain is meshed with a structured hexahedral grid, with a minimum cell height at the cylinder corresponding to $y^+ = yu_\tau/\nu = 1$, where $u_\tau = \sqrt{\tau_w/\rho}$ is the friction velocity at the wall. As shown by Fig. 3, the mesh has a O-grid topology, which is commonly used for flow around cylinders. For the present laminar flow computations, the cell height at the cylinder boundary of $\Delta y = 0.05$. A mesh convergence analysis was performed and led to the conclusions that a mesh with 30,096 cells was sufficient to obtain accurate converged results. The time step size was further varied to keep $CFL = 0.7$ for all the computations.

#### 3.1.2 Free vibration

This section shows the results of a cylinder vibrating freely in the direction transverse to the flow. Here, the flow is laminar and results are compared with those available in the literature (Carmo et al., 2011), where a loosely-coupled numerical FSI framework was also used. The simulations are conducted at a Reynolds number of $Re = 150$, a mass ratio $m^* = 4m/(\rho \pi D^2 L) = 2$ ($L$ being the length of the equivalent three-dimensional cylinder), and a damping factor of $\zeta = 0.007$. The reduced velocity $U^* = U/(\omega_n D)$ varies in the range $2.5 < U^* < 16$. The non-dimensional maximum amplitude of the cylinder is shown in Fig. 4, for both the present simulations and the results from the literature. The lock-in region is clearly identified for $4 < U^* < 7$ and good agreement between the present results and the reference solution is obtained for the whole range of reduced velocities. Figure 5 shows the cylinder frequency divided by its natural frequency. Again, good agreement is obtained between the present results and the literature. As expected, the motion frequency equals the natural frequency in the lock-in region (i.e. $\omega/\omega_n = 1$), whilst the frequency ratio follows the Strouhal relation outside the lock-in band. These results demonstrate that the present weakly-coupled FSI model is capable of predicting the dynamics of a light cylinder undergoing VIV in laminar flow conditions.

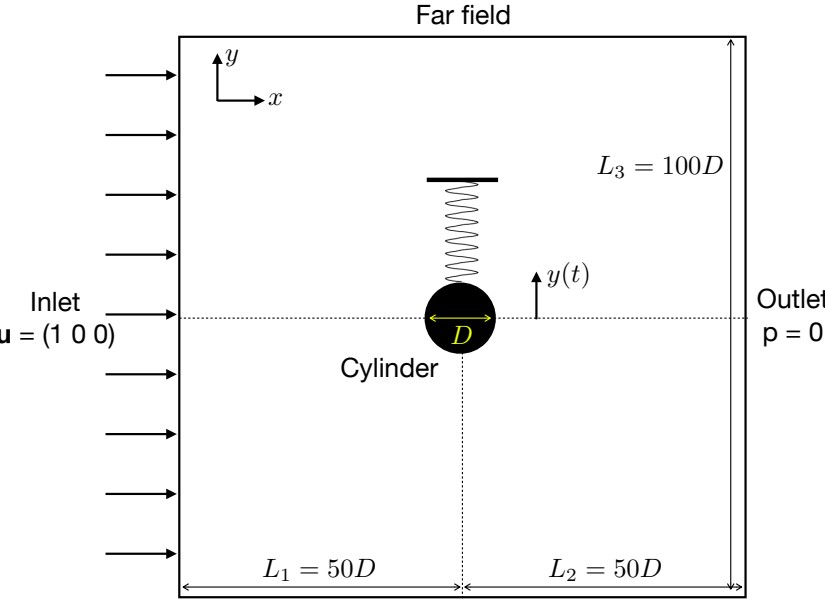

**Figure 2.** Sketch of the computational domain.

 ## 3.2 Turbulent flow

### 3.2.1 Simulation setup

The computational domain and boundary conditions are identical to those used in the previous subsection. However, since a turbulence model is used here, additional boundary and initial conditions are needed for the turbulent quantities. In particular, the initial value of the turbulent kinetic energy is set to

$$k_{t=0} = \frac{3}{2}\left(TIu_\infty\right)^2,$$ (19)

with the turbulence intensity being set at $TI = 0.03$ and the inlet velocity is $u_\infty = 1$, whilst the initial value of the turbulence specific dissipation rate is equal to

$$\omega_{t=0} = \frac{\rho k}{\mu}\left(\frac{\mu_t}{\mu}\right)^{-1},$$ (20)

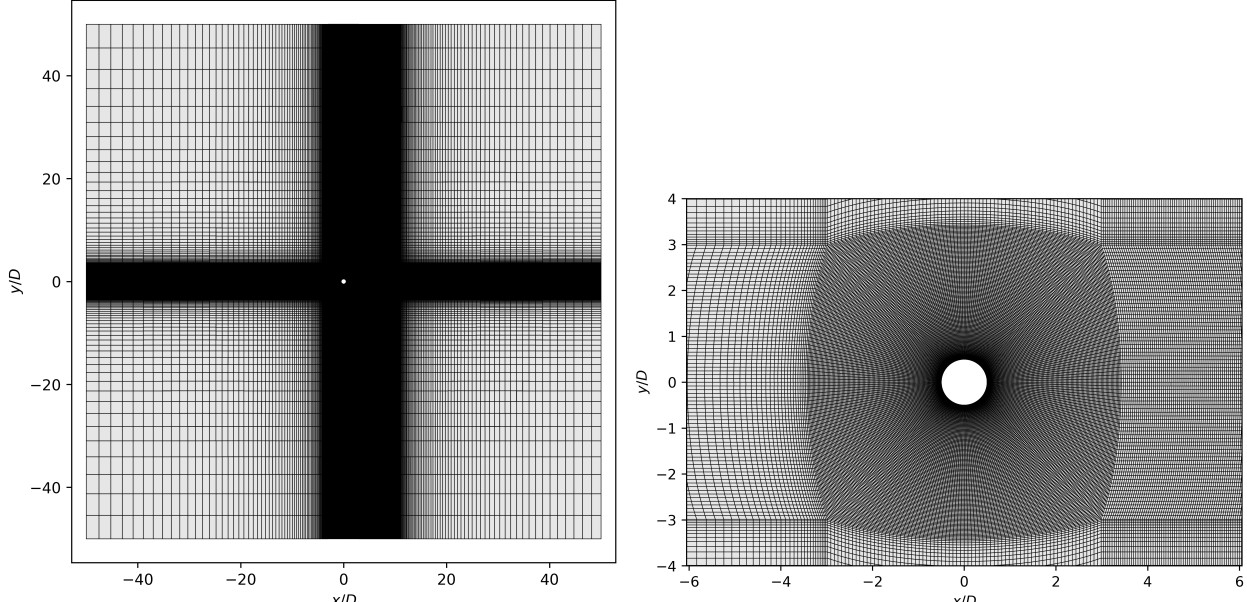

**Figure 3.** Computational mesh: full domain (left), zoom around the cylinder (right).

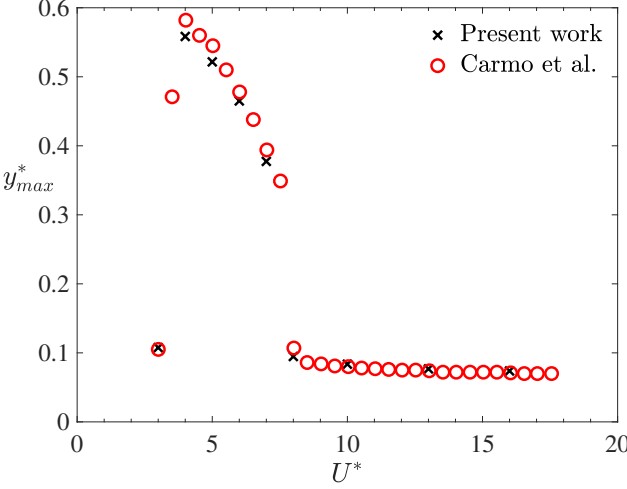

**Figure 4.** Non-dimensional maximum motion amplitude as a function of the reduced velocity for $Re = 150$, $m^* = 2$, and $\zeta = 0.007$.

with $\mu_t/\mu = 10$, $\mu$ being the fluid dynamic viscosity. The initial value of the eddy viscosity is $\nu_t = 0$. Regarding the boundary conditions, Dirichlet conditions are imposed on both $k$ and $\omega$ at the inlet, with values equal to the initial ones, respectively, and Neumann conditions are imposed at the outlet. At the cylinder, the mesh is fine enough to resolve the flow up to the viscous sublayer ($y^+ = 1$) using prism layers around the cylinder, with a mesh growth ratio equal to 1.2 and a minimum cell height at

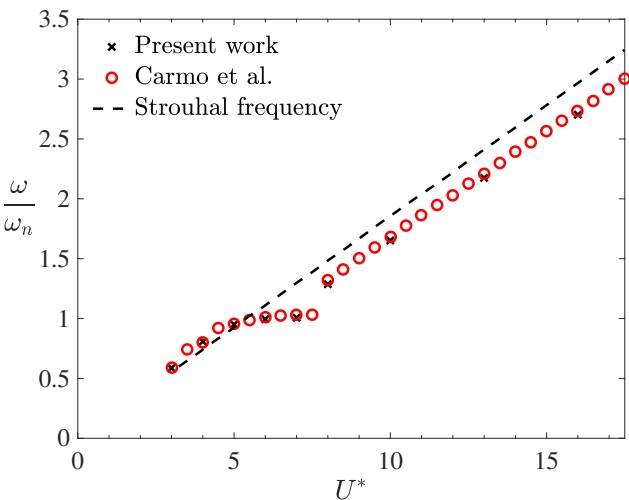

**Figure 5.** Ratio of the motion frequency to the natural frequency as a function of the reduced velocity for $Re = 150$, $m^* = 2$, and $\zeta = 0.007$.

the cylinder equal to $\Delta y = 7 \cdot 10^{-6}$. Converged results were obtained for a mesh with 98,730 grid points. As for the laminar flow simulations, the maximum CFL number was kept constant at $CFL = 0.7$. Since the viscous sublayer is mostly laminar, the turbulent quantities are such that turbulence is suppressed at the wall. Thus, at the cylinder boundary, $k = 10^{-10} m^2/s^2$, $\nu_t = 0$, and

$$\omega = \frac{6\mu}{\beta_1 y_{wall}^2}, \tag{21}$$

with $\beta_1 = 0.075$ and $y_{wall}$ being the mesh cell height at the wall (Menter, 1992).

### 3.2.2 Flow past a stationary cylinder

Before considering flow past moving cylinders, the accuracy of the $k - \omega$ SST turbulence model in the supercritical regime is first assessed on flow past a stationary cylinder at a Reynolds number $Re = 3.6 \cdot 10^6$. Table 2 compares the results of the present simulations with numerical and experimental data from the literature, in terms of Strouhal number $St$, mean drag coefficient $C_{D,\text{mean}}$, base pressure coefficient $c_{p,b}$ at $\theta = 180°$ ($\theta$ being the circumferential angle at a point on the cylinder surface, starting from the stagnation point), and flow separation angle $\theta_{\text{sep}}$. The values marked with a double asterisk are estimated indirectly based on the available pressure distribution where the constant pressure plateau is reached. Overall, it is observed that the present results agree well with the literature, and especially when compared with other numerical results obtained with either URANS (Ong et al., 2009) or DES (Travin et al., 2000) turbulence models. The pressure coefficient at the cylinder surface is further plotted in Fig. 6. Again, it is clear that the present URANS results match well the results from other numerical studies. Compared to experimental works (Achenbach, 1968; Jones et al., 1969), the present pressure coefficient agrees well in the front part of the cylinder, for $0° \leq \theta \leq 63°$. For larger values of $\theta$, the numerical results show both an earlier pressure recovery (at around $\theta \approx 80°$) and a less pronounced negative peak for $c_p$. It was shown by Achenbach (1968) that the transition point of the

**Table 2.** Comparison of data for flow past a stationary cylinder in the supercritical regime.

| Data | $Re$ | $St$ | $C_{D,\text{mean}}$ | $c_{p,b}$ | $\theta_{\text{sep}}$ (°) |
|---|---|---|---|---|---|
| Present work | $3.6 \cdot 10^6$ | 0.32 | 0.42 | -0.5 | 111 |
| 2D k-$\epsilon$ URANS (Ong et al., 2009) | $3.6 \cdot 10^6$ | 0.31 | 0.46 | $\approx$ -0.54[**] | 114 |
| DES (Travin et al., 2000) | $3 \cdot 10^6$ | 0.35 | 0.41 | -0.53 | 111 |
| Experimental (Achenbach, 1968) | $3.6 \cdot 10^6$ | 0.25 | 0.76 | $\approx$ -0.81[**] | 115 |
| Experimental (Jones et al., 1969) | $3.5 \cdot 10^6$ | 0.24 | 0.58 | $\approx$ -0.65[**] | $\approx$ 125[**] |

boundary layer is at $\theta = 65°$ for that Reynolds number. It seems that the pressure distribution starts to deviate around this point, which could indicate that the transition has an impact on the pressure distribution. The boundary layer in the experiment of Jones et al. (1969) continues to decelerate further aft (at $\theta = 90°$), which explains why the separation also occurs later in their measurements. This might indicate that the numerical models do not fully predict the detached flow accurately. This in line with previous observations made in the literature, although at different Reynolds numbers (Ferrand et al., 2006; Catalano et al., 2003). The friction coefficient at the cylinder surface is also plotted in Fig. 7. The present numerical results agree well with other numerical studies that use either LES (Catalano et al., 2003) or DES (Travin et al., 2000). The peak of maximum friction coefficient slightly underestimates the other numerical studies. However, the overall agreement is good and the separation location is also accurately captured and compares well with the experiment of Achenbach (1968). The numerical values of the friction coefficient $c_f$ for $0° \leq \theta \leq 90°$ are generally much higher than the experimental values. This can be explained by the fact that the numerical studies assume fully turbulent transition, and therefore, lead to significantly higher values of friction coefficient. In particular, the over-prediction is caused by the fact that the laminar flow at the beginning part of the cylinder wall is not taken into account in the numerical models. This is in line with the observations of Travin et al. (2000) and Catalano et al. (2003). Only the URANS simulations of Ong et al. (2009) show much smaller values of friction coefficient. However, this study used a $k-\epsilon$ turbulence model, which is known to perform poorly with flow separation and strong pressure gradients. This could explain why these URANS results under-predict the skin-friction coefficient compared to the other numerical studies. Despite the discrepancies in $c_p$ and $c_f$ between numerical and experimental results, the present turbulence model is deemed to be adequate for the purpose of this work as it provides results that are in line with other trustful numerical studies in the supercritical regime.

### 3.2.3 Flow past a cylinder under forced vibration

In this section, the transverse motion of the cylinder is prescribed for different frequencies $\omega_f$. Seven different frequencies are considered, each corresponding to a different reduced velocity and Reynolds number, as indicated in Table 3. In the latter, $St_f$ denotes the non-dimensional forcing frequency of the cylinder, while $St$ is the Strouhal number associated with flow past a stationary cylinder at the same Reynolds number. The latter is obtained numerically with the same mesh as for the corresponding moving cases. The simulations are further ran for seven different values of the prescribed maximum motion

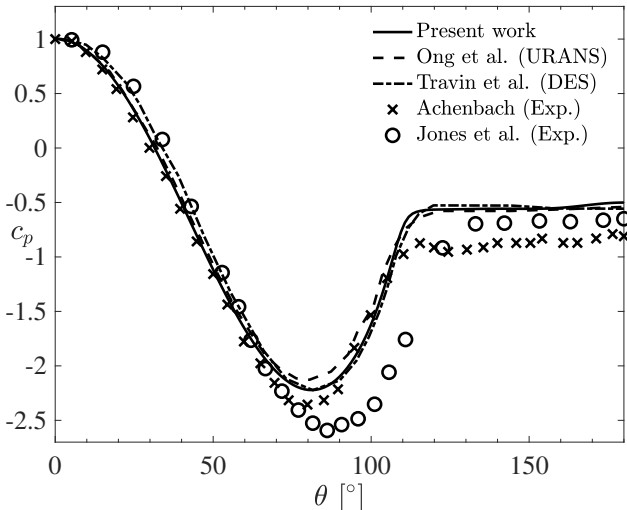

**Figure 6.** Time-averaged pressure distribution for flow past a stationary cylinder in the supercritical turbulent regime.

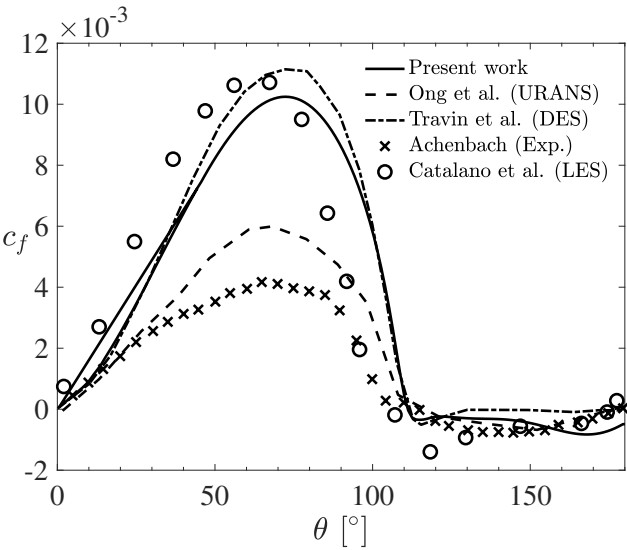

**Figure 7.** Time-averaged friction coefficient for flow past a stationary cylinder in the supercritical turbulent regime.

amplitude. The root-mean-square of the lift coefficient (non-dimensionalised by its value for a stationary cylinder) is shown in Fig. 8, for different ratios $St_f/St$ and prescribed motion amplitudes $y^*_{max}$. As expected, the lift magnification due to the cylinder motion increases with both the non-dimensional frequencies and $y^*_{max}$. The increase with frequency is rather linear, except close to $St_f/St = 1$. The peak in lift magnification around a unitary reduced frequency is believed to be a consequence of the wake lock-in on the cylinder motion. For $St_f/St > 1$ and $y^*_{max} \geq 1$, our simulation results indicate that

**Table 3.** Range of frequency ratios and velocities considered for the forced vibration of a cylinder in the supercritical regime.

| $St_f/St$ | $U^*$ | $Re$ |
|---|---|---|
| 0.58 | 2.524 | $3.8 \cdot 10^6$ |
| 0.74 | 2.869 | $4.3 \cdot 10^6$ |
| 0.89 | 3 | $4.5 \cdot 10^6$ |
| 0.95 | 3.223 | $4.8 \cdot 10^6$ |
| 1.01 | 3.384 | $5 \cdot 10^6$ |
| 1.07 | 4 | $6 \cdot 10^6$ |
| 1.23 | 5 | $7 \cdot 10^6$ |

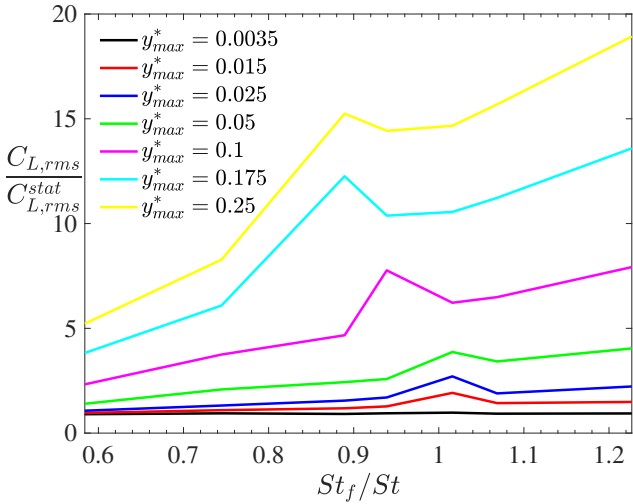

**Figure 8.** Root-mean-square of the lift coefficient (non-dimensionalised by its value for a stationary cylinder) for forced vibrations in the supercritical turbulent regime with different frequency ratios $St_f/St$ and prescribed motion amplitudes $y^*_{max}$.

the lift magnification becomes very large, which is believed to be due to the effective added mass of the fluid caused by the vorticity dynamics (Williamson and Govardhan, 2008).

Another observation is that the lock-in band width increases with increasing motion amplitudes, with a move towards lower reduced frequencies as the oscillation amplitude increases. Both these observations, namely increased lift magnification at high frequency ratios and wider lock-in band at larger motions, were observed in the experimental results obtained from NASA under similar conditions (Jones et al., 1969), as shown in Fig. 9. Although the experimental results show some scatter, a peak of lift magnification is obtained around $St_f/St = 1$, decreasing for slightly larger ratios of Strouhal number, before increasing further as $St_f/St > 1$ (see for example the results at $y^*_{max} = 0.0278$ in Fig. 9).

Although the cylinder motion is prescribed, it is interesting to look in more details at the feedback mechanism between the cylinder and the fluid motions. In order to do this, the aerodynamic damping is analysed. It is defined as the part of the

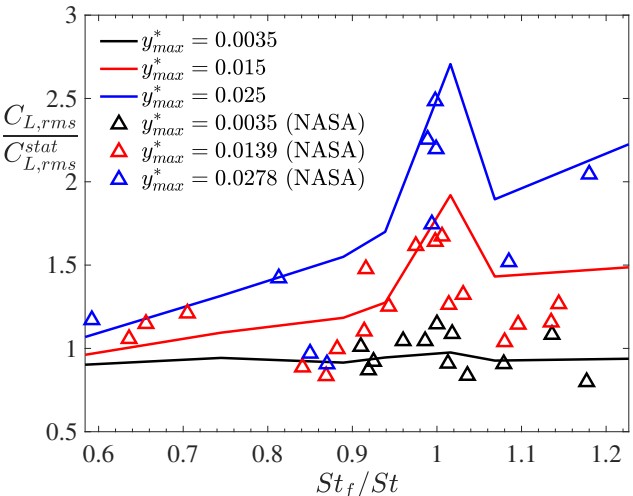

**Figure 9.** Root-mean-square of the lift coefficient (non-dimensionalised by its value for a stationary cylinder) for forced vibrations compared with NASA experimental data (Jones et al., 1969) at three values of the prescribed motion amplitude.

harmonic lift force that is in phase with the cylinder velocity. There are different ways to compute this damping. Here, an approach based on the time-averaged energy transfer between fluid and vibrating cylinder is considered (Bourguet et al., 2011; Gopalakrishnan, 1993), as it is believed to be accurate when signals exhibit multiple frequencies. This leads to the following
expression for the aerodynamic damping,

$$C_{L,V} = -\sqrt{2}\frac{\overline{C_L' \dot{y}'}}{\sqrt{\overline{\dot{y}^2}}}, \tag{22}$$

where $C_L = 2F_y/(\rho U^2 D)$ denotes the lift coefficient, a prime denotes the fluctuating part of a quantity and an over-bar denotes its time average. The aerodynamic damping is plotted for the different simulation cases in Fig. 10. For $St_f/St \geq 1$, it is observed that $C_{L,V}$ increases with the motion amplitude. This indicates that the wake dynamics tends to oppose the cylin-
der motion as the latter increases, hence stabilising the system. By contrast, for $St_f/St \leq 0.75$, the aerodynamic damping is negative with an absolute value increasing as the cylinder displacement increases. In that case, the wake dynamics tends to further amplify the system motion, hence leading to an unstable system behaviour. In the lock-in band, for $0.8 \leq St_f/St \leq 1$, the nature of the interaction between fluid and structural motion depends on the amplitude of the prescribed motion. However, a stabilising fluid-structure interaction behaviour is observed for large motion amplitudes. These observations agree with
that obtained from Jones et al. (1969). In particular, the overall behaviour of the experimental aerodynamic damping in the supercritical turbulent regime, for varying frequency ratios, was found to be similar to the present CFD results.

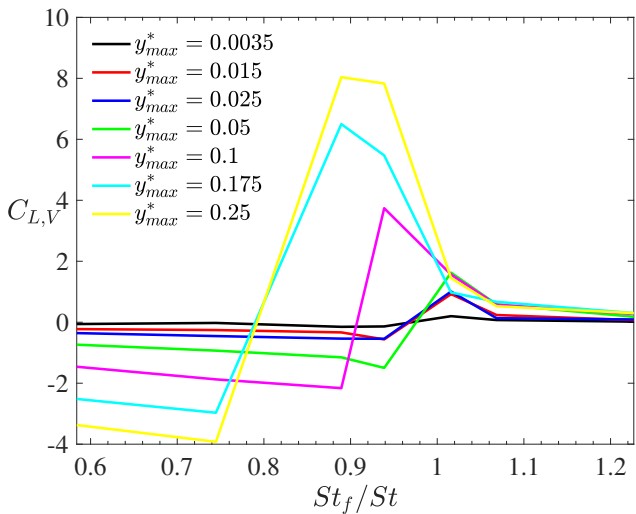

**Figure 10.** Aerodynamic damping for forced vibrations in the supercritical turbulent regime with different frequency ratios $St_f/St$ and prescribed motion amplitudes $y^*_{max}$.

### 3.2.4 Flow past a freely-vibrating cylinder

In this section, the cylinder is left to freely vibrate under the vortex-induced vibrations. The results are shown for a baseline value of mass ratio $m^* = 29.6$ and damping coefficient $\zeta = 0.003$, which are representative of a realistic wind turbine tower. The effect of these values on the results is also briefly discussed in this section. The reduced velocity is varied from $U^* = 1.88$ to $U^* = 10.74$, corresponding to Reynolds numbers between $Re = 2.8 \cdot 10^6$ and $Re = 1.6 \cdot 10^7$, respectively. The associated non-dimensional maximum motion amplitude under these conditions is shown in Fig. 11, in which the lock-in band can be clearly visualised. Additionally, the insets show the time evolution of the lift coefficient on the cylinder for certain values of reduced velocity. The associated values of reduced frequency, $\omega^* = \omega_n/\omega_{(\text{shed, stat})}$, are also shown for each inset. These insets illustrate that for some values of $U^*$, a non-harmonic lift response (and consequently also motion response) is obtained. Outside the lock-in region, at low values of $U^*$, the vortex shedding frequency is dominated by the stationary Strouhal relation and is very different from the natural frequency of the cylinder. This is also apparent from Fig. 12. In that case, the mean aerodynamic forces are similar to those of a stationary cylinder and little cylinder motion is observed. Furthermore, the vorticity contours in the cylinder wake follow a 2S pattern, similarly to flow past a stationary cylinder. Also, the phase angle $\phi$ between the lift force and the transverse displacement is constant and equals $0°$. When $U^*$ increases to values close to the lock-in band, the time histories of the aerodynamic forces and cylinder motion change. First, the cylinder displacement increases substantially because the vortex shedding frequency gets closer to the natural frequency of the cylinder. Second, the cylinder motion shows two main frequency responses instead of one. The dominant frequency in the time evolutions of the lift and motion amplitude still equals the Strouhal frequency. However, performing a power spectral density on these time signals shows that a second (much smaller) frequency peak coincides with the natural frequency. This explains the observed increase in cylinder motion compared to cases

at smaller reduced velocities. The phase angle between lift force and transverse displacement is also different than for smaller values of $U^*$. Instead of being constant, it periodically oscillates in time around a mean value of $\phi = 5°$. When the phase angle is larger than the mean value, the aerodynamic damping decreases and the fluid amplifies the cylinder excitation. By contrast, when the phase angle is smaller than the mean value, the fluid damps the cylinder motion. When $U^*$ is increased further, inside the lock-in band, the time histories of the lift and displacement also show two frequencies. However, the dominant peak corresponds to the natural frequency of the cylinder, hence confirming that the wake behaviour becomes driven by the cylinder oscillations rather than the Strouhal relation. Furthermore, the lift force (and also motion displacement) alternate between growth and decay, which also corresponds to a change in sign of the aerodynamic damping. In particular, when the lift force increases, the fluid dynamics excites the cylinder. By contrast, when the lift force decreases, the fluid dynamics damps the cylinder motion. As such, the vortex-induced vibrations continuously alternate between self-exciting and self-limiting behaviours. This alternation can also be related to a drop in phase angle between lift force and cylinder displacement. Figure 13 shows the time evolution of the phase angle for $U^* = 3.1$. It is apparent that the phase angle changes sign as time evolves. When $\phi > 0$, the fluid excites the cylinder motion and the magnitude of both the lift force and the transverse displacement increases. By contrast, when $\phi < 0$, the lift force lags behind the displacement and the fluid dampens the cylinder motion. The magnitude of both the lift force and the cylinder displacement also decreases. The phase angle drop can also be related to the wake pattern development as shown in Fig. 14. The left figure corresponds to a positive phase angle at $t/T_s = 143.5$, while the right figure is taken at $t/T_s = 146.5$, when the phase angle is negative. When the phase angle is positive, the vortices are shed from the bottom side of the cylinder when the cylinder reaches its maximum positive displacement. The opposite is true when the phase angle switches sign. In that case, the vortices are shed from the same side as where the cylinder is oscillating to. It seems that, when the cylinder motion becomes too large, the wake reorganises itself in such a way that it starts damping the cylinder motion. The self-exciting behaviour becomes a self-limiting behaviour. This alternation of behaviours is influenced by both the mass ratio and the damping factor. If the mass ratio is reduced, the non-dimensional displacement amplitude increases. This is expected as a lower mass ratio leads to a lower structural inertia when compared to the fluid inertia. This makes the cylinder more susceptible to oscillations, and possibly lock-in. By contrast, for very large mass ratios, the cylinder motion decreases and eventually tends to a stationary cylinder. There is thus a critical value of the mass ratio at which the system stops undergoing enhanced fluid-structure interactions. Non-harmonic behaviours of the lift and motion displacement are also less likely to occur at large mass ratios and large damping factors because the larger structural inertia prevents the cylinder from changing to a different dynamic state. When $U^*$ is increased further, in a certain range, the lift force, drag force and transverse displacement do not show a clear converged pattern (see inset in Fig. 11 at $U^* = 4.3$). It is not excluded that, if the simulation was run for even longer times, a self-limiting behaviour could appear. However, the observed behaviour is believed to have physical causes, rather than being a numerical artifact. This is because the operation points for which this happens correspond to reduced frequencies between $0.5 \leq \omega^* \leq 0.7$, for which the results of the forced-vibration cases led to unstable behaviours. By contrast, at smaller values of $U^*$, the reduced frequencies are large ($0.9 \leq \omega^* \leq 1.2$) and a stable forced-vibration behaviour was obtained. It is also worth noting that the large lock-in band observed in the present study is in line with previous observations made in the literature at similar conditions. For example, Guilmineau and Queutey (2004) and

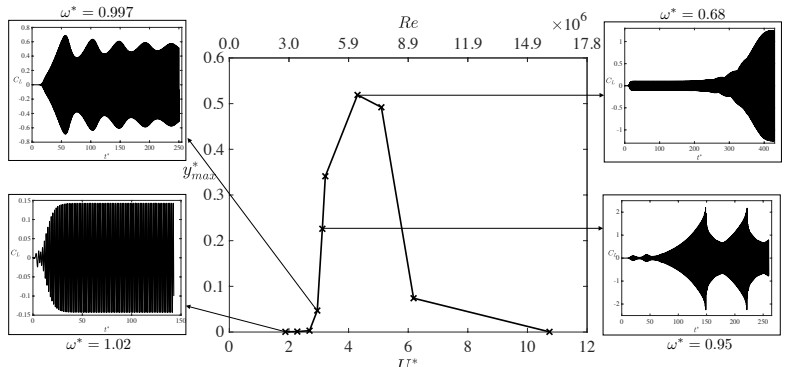

**Figure 11.** Non-dimensional maximum motion amplitude as a function of the reduced velocity for $m^* = 29.6$, $\zeta = 0.003$ and $1.88 \leq U^* \leq 10.74$. Insets show the time evolution of the lift coefficient at certain values of reduced velocities.

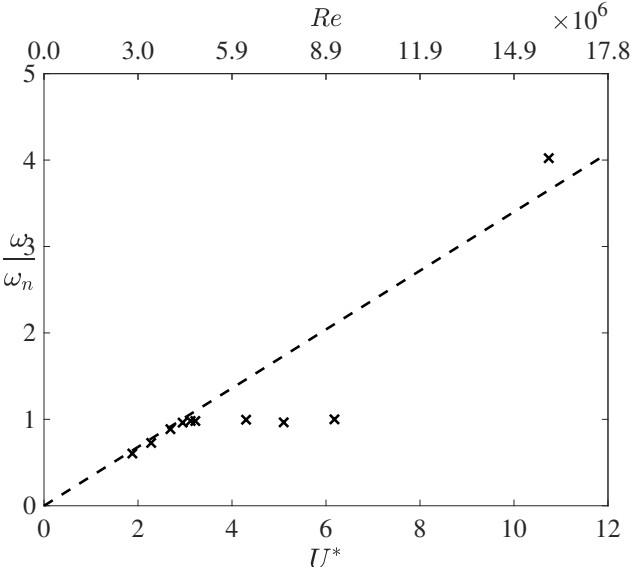

**Figure 12.** Ratio of the motion frequency to the natural frequency as a function of the reduced velocity for $m^* = 29.6$, $\zeta = 0.003$ and $1.88 \leq U^* \leq 10.74$.

Assi (2009) showed large values of non-dimensional displacement for reduced frequencies as low as $\omega^* = 0.7$, whilst the wind tunnel experiment of Feng (1968) also showed a lock-in region extending to a frequency ratio of $\omega^* = 0.6$.

## 4   Conclusions

In this work, numerical simulations were performed of a two-dimensional cylinder undergoing vortex-induced vibrations. First, 340  the fluid-structure interaction methodology was validated in the laminar regime, for which direct numerical simulations (DNS)

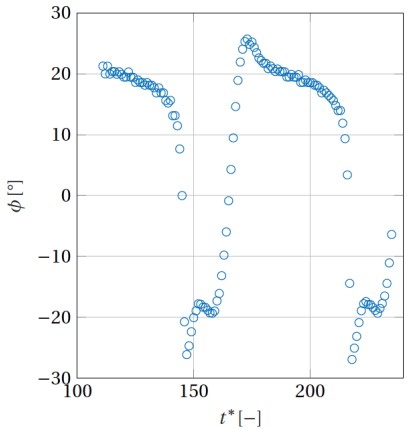

**Figure 13.** Time evolution of the phase angle between lift force and transverse displacement at $U^* = 3.1$. Figure taken from Derksen (2019).

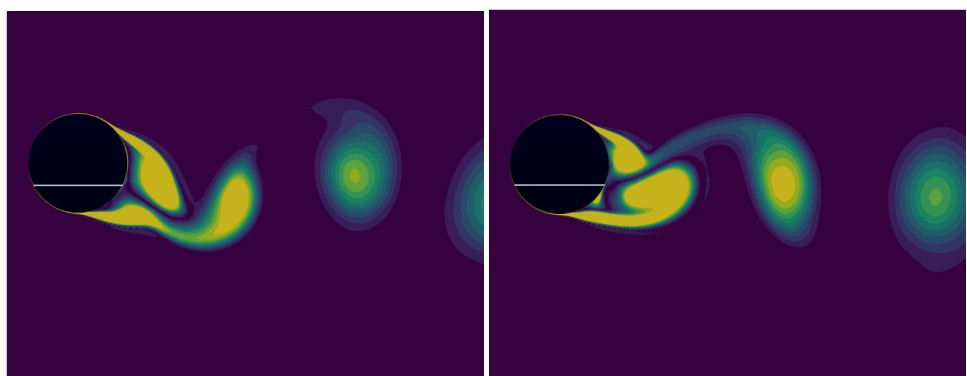

**Figure 14.** Instantaneous non-dimensional vorticity magnitude for $U^* = 3.1$ at the maximum positive displacement: $t/T_s = 143.5$ (left) and $t/T_s = 146.5$ (right). The vorticity is filtered to the range $0 \leq \xi D/U_\infty \leq 5$. Figures taken from Derksen (2019).

were able to reproduce results from the literature. Second, supercritical turbulent regimes experienced by wind turbine towers were simulated using the $k - \omega$ SST unsteady Reynolds-Averaged Navier–Stokes (URANS) model. The ability of our model to simulate flow past a stationary cylinder in the turbulent supercritical regime was demonstrated by comparing the present results with other URANS, large-eddy simulations (LES), detached-eddy simulations (DES) and experimental results from the
literature. Additionally, the present tools were capable of simulating the dynamic behaviour of the cylinder (including lock-in) for a range of reduced velocities and Reynolds numbers. In particular, good agreement was found between the present results and those obtained from wind tunnel experiments under forced vibrations. When the cylinder was left free to oscillate under the effect of vortex shedding, the results highlighted a complex interplay between structural and fluid dynamics for values of reduced velocity close to, or inside, the lock-in band. In particular, there was a continuous alternation between self-exciting
and self-limiting vortex-induced vibrations. The physical feasibility of this interplay was supported by both the role of the

aerodynamic damping, which was shown to continuously change sign at these operating points, and the results from forced-vibration oscillations under similar conditions.

Traditionally, the designs of wind turbine towers are soft-stiff, with a tower natural frequency larger than the rotor rotational frequency but smaller than the blade passing frequency. Today, advanced wind turbine controllers can work through resonance conditions, allowing the tower natural frequencies to overlap with the rotor rotational frequency and the blade passing frequency. This makes soft-soft designs possible, which is promising to enable a significant reduction of tower mass when the tower height is larger than a hundred metres (Dykes et al., 2018). During the installation phase of wind turbines, however, the rotor-nacelle assembly is absent and wind turbine controllers cannot be used to damp vibrations. This is the case during the whole installation cycle of jackets or monopile foundations, as well as their vertical storage and transport. During that time, towers might stand without rotors for several weeks. They act as a clamped beam and are subjected to supercritical Reynolds numbers. Over the past years, the tower clamped first mode frequency has been reduced and this trend is continuing. This means that both the combined first eigenmodes and the second mode of vibration can interact with the wind climate and become critical for the dynamics of the structure and VIV. The present work shows that, in the supercritical flow regime, a cylinder can experience episodes of sustained VIV, where the Strouhal relation is not valid. This means that the best mitigation strategy is to have a robust damping mechanism that is effective for the range of critical first bending frequencies. This work presents some limitations that should be addressed in future study. For example, future work should look at three-dimensional tower sections and also the influence of non-uniform wind flows on the present results. Although this has already been partly investigated in the literature, analyses lack for the supercritical regime. Also, the use of a tapered cylinder instead of a circular one is expected to lower the risk of VIV and move the lock-in region to lower reduced velocities. Nevertheless, circular cylinders are very relevant to new wind turbine tower designs, as some manufacturers are currently looking into having minimal taper at the top sections. Finally, during the installation phase, wind turbine towers are usually transported in group. Therefore, wake-tower interactions and their effect on the development of VIV should be further investigated.

*Acknowledgements.* Viré acknowledges support from the European Commission under the H2020 ITN project STEP4WIND (grant agreement no. 860737) funded by the European Union's Horizon 2020 research and innovation programme under the Marie Skłodowska-Curie scheme. During the research, Folkersma was supported by the H2020 ITN project AWESCO (grant agreement no. 642682). The opinions expressed in this document reflect only the author's view and in no way reflect the European Commission's opinions. The European Commission is not responsible for any use that may be made of the information it contains.

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
