# Peer review of "Two-dimensional numerical simulations of vortex-induced vibrations for a cylinder in conditions representative of wind turbine towers"

_Wind Energy Science, 2019_

## Referee Comment (RC1) · Anonymous Referee #1 · 9 Jan 2020

\*\*\*\*\*Summary\*\*\*\*\*

The authors present a set of numerical simulations of a 2D-cylinder, which is mounted to an elastic system an allowed to oscillate transversely. This problem has been widely studied in the literature by means of both numerical methods and experiments, and it constitutes the canonic problem of Vortex Induced Vibrations (VIV).

The followed methodology is properly documented, as well as the choices made while building the different FSI set-ups. As a preliminary step, the authors also compare the results of their numerical methods with those issue from other simulations and experiments. The analysis of the results is very detailed and didactic.

The set-up of the carried out simulations is representative of the structural properties and inflow conditions of wind turbine towers. This is indeed the added value of this work, since no previous experiences in these lines were performed.

The paper is very well organized and clearly written.

Below some remarks in order to improve the manuscript.

*****Requested modifications*****

++ General ++

Some of the variables used in the text are not properly introduced. I suggest to include an acronym section in the manuscript, or alternatively to make sure that all the symbols are properly described.

++ Title ++

The title could be misleading, since no wind turbine tower is modeled in this work. I would suggest to make it more explicit, stating that a 2-D cylinder was simulated (in conditions that are representative of wind turbine towers). It should be mentioned that the content of the paper is very clear regarding this point later on.

++ Abstract ++

Change "verified by considering" by "validated by considering". It is a more appropriate word in this context.

++ Introduction ++

Line #28: I think you should be more cautious with the statement "Although wind turbines towers are tapered, they resemble a circular cylinder". The tapering of the tower may indeed have non-negligible consequences on the VIV phenomenon, and in addition wind shear can be also present. This will introduce more frequencies into play, making the VIV phenomenon much more complex [Balasubramanian et. al (1998), Balasubramanian et. Al (2001), Bourguet et .al (2011), Bourguet et .al (2013) , Hover et. al (1998)]. So someone might think that the proper way to simulate the VIV of towers could be to perform "a series of 2D simulations" corresponding to the inflow conditions at different heights (as presented in WESC 2019). In addition, the spring-based structural model employed in this work may also present some limitations when compared to the 3D structure.

Line #20: It is interesting to emphasize if the four works mentioned here were based on computational studies and/or experiments.

++ Section 2.2 ++

More details about the "modal condensation" of the tower should be provided. - What model of tower is being represented?. Could you provide some dimensions and structural properties?. Will the conclusions of this work hold for other towers? - More details about how the modal analysis is performed and the passage to a 2D geometry are necessary.

*****Recommendations*****

I missed a Figure showing the mesh, even if it is described in the document. It could be useful to understand how it expands from the wall surfaces to the inner CFD domain. Eventually you could show how the mesh is deformed for the cases with important cylinder displacements.

There is not comment regarding the wakes. This could be interesting, probably in combination with the discussion of the freely oscillating cylinder.

*****Comments*****

The discussion concerning the "St" vs. "cylinder natural frequency" competition (page #15) reminded me of the study of Bourguet and Triantafyllou (2016), that is also a very relevant work.

*****References*****

Balasubramanian, S., Haan, F. L., Szewczyk, A. A., & Skop, R. A. (1998). On the existence of a critical shear parameter for cellular vortex shedding from cylinders in nonuniform flow. Journal of Fluids and Structures, 12(1), 3–15. https://doi.org/10.1006/jfls.1997.0122

Balasubramanian, S., Haan, F. L., Szewczyk, A. A., & Skop, R. A. (2001). An experimental investigation of the vortex-excited vibrations of pivoted tapered circular cylinders in uniform and shear flow. Journal of Wind Engineering and Industrial Aerodynamics, 89(9), 757–784. https://doi.org/10.1016/S0167-6105(00)00093-3

Hover, F. S., Techet, A. H., & Triantafyllou, M. S. (1998). Forces on oscillating uniform and tapered cylinders in crossflow. Journal of Fluid Mechanics, 363, 97–114. https://doi.org/10.1017/S0022112098001074

Bourguet, R., Modarres-Sadeghi, Y., Karniadakis, G. E., & Triantafyllou, M. S. (2011). Wake-body resonance of long flexible structures is dominated by counterclockwise orbits. Physical Review Letters, 107(13), 1–4. https://doi.org/10.1103/PhysRevLett.107.134502

Bourguet, R., Karniadakis, G. E., & Triantafyllou, M. S. (2013). Distributed lock-in drives broadband vortex-induced vibrations of a long flexible cylinder in shear flow. Journal of Fluid Mechanics, 717, 361–375. https://doi.org/10.1017/jfm.2012.576

Bourguet, R., & Triantafyllou, M. (2016). The onset of vortex-induced vibrations of a flexible cylinder at large inclination angle. Journal of Fluid Mechanics, 809, 111-134. doi:10.1017/jfm.2016.657

---

## Referee Comment (RC2) · Anonymous Referee #2 · 7 Feb 2020

The article investigates a relevant topic of VIV over tall towers of wind turbines during the installation process. However several points are not explained clearly. The article should also explain the following points before publication: 1) Introduction, Page 1: Provide some numbers as to what you consider as a large diameter or tall wind turbine for which VIV is relevant. 2) Section 2.1 - While you outline the k-w CFD model in detail, there is not much mentioned about the external wind conditions. What free wind condition range is applicable to your model in terms of free wind speed and turbulence? Can you also consider wind shear and ground effect? 3) Page 5: Why is there no aerodynamic damping term present in Eq. (18)? 4) Page 7: From figure 2, it appears that bending in only one direction is considered. For a wind turbine tower, both sideside and fore-aft modes are excited in VIV and so at least two springs in perpendicular directions should be considered. 5) Section 3:0 : Can you provide a figure of the CFD mesh you used? 6) Page 8: You state the structural damping is 0.007. How much artificial damping due to the CFD mesh do you generate in your model? Is this artificial damping significant with respect to your structural damping? 7) Same question for the turbulent flow: How much artificial/numerical damping is present in your CFD model and what affect does that have on the results? 8) Going by the results of Table 2 on the angle of flow separation over the cylinder, what would be the best direction to orient the spring for your structural oscillation, since you consider only single dimensional oscillations? 9) Page 12, line 240: The expression for aerodynamic damping used is not clear. There is no 'q' term in the equation as given in the explanation. 10) Figure 7, Figure 8 etc: Can you also plot this versus the Strouhal number? 11) Section 3.2.4: When you state realistic wind turbine tower, what is the wind turbine tower diameter, height and natural frequency that is considered? 12) Can you conclude on how the results of your work can be applied to an existing wind turbine tower? What wind conditions and tower natural frequencies should the turbine designer pay attention to for VIV?

---

## Author Comment (AC1) · 5 Mar 2020

The authors thank the Reviewer for their detailed and positive feedback. We have taken the feedback into account in the revised manuscript. Changes are highlighted in red. Each comment is further addressed below.

++ General ++ Referee: Some of the variables used in the text are not properly introduced. I suggest to include an acronym section in the manuscript, or alternatively to make sure that all the symbols are properly described.

Response: Thanks for pointing this out. We scanned the whole manuscript and edited

the text to define all the variables that were not properly introduced.

++ Title ++ Referee: The title could be misleading, since no wind turbine tower is modeled in this work. I would suggest to make it more explicit, stating that a 2-D cylinder was simulated (in conditions that are representative of wind turbine towers). It should be mentioned that the content of the paper is very clear regarding this point later on.

Response: The title has been changed accordingly.

++ Abstract ++ Referee: Change "verified by considering" by "validated by considering". It is a more appropriate word in this context.

Response: The wording has been changed accordingly.

++ Introduction ++ Referee: Line #28: I think you should be more cautious with the statement "Although wind turbines towers are tapered, they resemble a circular cylinder". The tapering of the tower may indeed have non-negligible consequences on the VIV phenomenon, and in addition wind shear can be also present. This will introduce more frequencies into play, making the VIV phenomenon much more complex [Balasubramanian et. al (1998), Bal-asubramanian et. Al (2001), Bourguet et .al (2011), Bourguet et .al (2013) , Hover et. al (1998)]. So someone might think that the proper way to simulate the VIV of towers could be to perform "a series of 2D simulations" corresponding to the inflow conditions at different heights (as presented in WESC 2019). In addition, the spring-based structural model employed in this work may also present some limitations when compared to the 3D structure.

Response: We agree that our statement could be misleading. We have therefore changed it and also added some references.

Referee: Line #20: It is interesting to emphasize if the four works mentioned here were based on computational studies and/or experiments.

Response: This is clarified in the text.

++ Section 2.2 ++ Referee: More details about the "modal condensation" of the tower should be provided. - What model of tower is being represented?. Could you provide some dimensions and structural properties?. Will the conclusions of this work hold for other towers? - More details about how the modal analysis is performed and the passage to a 2D geometry are necessary.

Response: We now provide more information about this in the text. The three-dimensional tower is divided into 40 segments with 6 degrees-of-freedom at each node, leading to stiffness and mass matrices of dimensions 246x246. Solving the system K-omegaˆ2*M=0 returns the natural frequencies of all the modes and the eigenvectors. Modal mass/stiffness is obtained by pre-multiplying and post-multiplying the mass and stiffness matrix by the eigenvector of the first bending mode. The procedure should be generic enough so that it also holds for other towers.

*****Recommendations***** Referee: I missed a Figure showing the mesh, even if it is described in the document. It could be useful to understand how it expands from the wall surfaces to the inner CFD domain. Eventually you could show how the mesh is deformed for the cases with important cylinder displacements.

Response: We added 2 figures to show the mesh topology.

Referee: There is not comment regarding the wakes. This could be interesting, probably in combination with the discussion of the freely oscillating cylinder.

Response: We added some discussion on the phase angle and vortex patterns for the freely-oscillating cylinder.

*****Comments***** Referee: The discussion concerning the "St" vs. "cylinder natural frequency" competition (page #15) reminded me of the study of Bourguet and Triantafyllou (2016), that is also a very relevant work.

Response: Thank you very much for pointing this out.

---

## Author Response (AR1)

**Response to Reviewer 1**

The authors thank the Reviewer for their detailed and positive feedback. We have taken the feedback into account in the revised manuscript. Changes are highlighted in red. Each comment is further addressed below.

++ General ++

*Some of the variables used in the text are not properly introduced. I suggest to include an acronym section in the manuscript, or alternatively to make sure that all the symbols are properly described.*

Thanks for pointing this out. We scanned the whole manuscript and edited the text to define all the variables that were not properly introduced.

++ Title ++

The title could be misleading, since no wind turbine tower is modeled in this work. I would suggest to make it more explicit, stating that a 2-D cylinder was simulated (in conditions that are representative of wind turbine towers). It should be mentioned that the content of the paper is very clear regarding this point later on.

The title has been changed accordingly.

++ Abstract ++

Change "verified by considering" by "validated by considering". It is a more appropriate word in this context.

The wording has been changed accordingly.

++ Introduction ++

Line #28: I think you should be more cautious with the statement "Although wind turbines towers are tapered, they resemble a circular cylinder". The tapering of the tower may indeed have non-negligible consequences on the VIV phenomenon, and in addition wind shear can be also present. This will introduce more frequencies into play, making the VIV phenomenon much more complex [Balasubramanian et. al (1998), Balasubramanian et. Al (2001), Bourguet et .al (2011), Bourguet et .al (2013) , Hover et. al (1998)]. So someone might think that the proper way to simulate the VIV of towers could be to perform "a series of 2D simulations" corresponding to the inflow conditions at different heights (as presented in WESC 2019). In addition, the spring-based structural model employed in this work may also present some limitations when compared to the 3D structure.

We agree that our statement could be misleading. We have therefore changed it and also added some references.

Line #20: It is interesting to emphasize if the four works mentioned here were based on computational studies and/or experiments.

This is clarified in the text.

++ Section 2.2 ++
More details about the "modal condensation" of the tower should be provided. - What model of tower is being represented?. Could you provide some dimensions and structural properties?. Will the conclusions of this work hold for other towers? - More details about how the modal analysis is performed and the passage to a 2D geometry are necessary.

We now provide more information about this in the text. The three-dimensional tower is divided into 40 segments with 6 degrees-of-freedom at each node, leading to stiffness and mass matrices of dimensions 246x246. Solving the system K-omega^2*M=0 returns the natural frequencies of all the modes and the eigenvectors. Modal mass/stiffness is obtained by pre-multiplying and post-multiplying the mass and stiffness matrix by the eigenvector of the first bending mode. The procedure should be generic enough so that it also holds for other towers.

*****Recommendations*****
I missed a Figure showing the mesh, even if it is described in the document. It could be useful to understand how it expands from the wall surfaces to the inner CFD domain. Eventually you could show how the mesh is deformed for the cases with important cylinder displacements.

We added 2 figures to show the mesh topology.

There is not comment regarding the wakes. This could be interesting, probably in combination with the discussion of the freely oscillating cylinder.

We added some discussion on the phase angle and vortex patterns for the freely-oscillating cylinder.

*****Comments*****
The discussion concerning the "St" vs. "cylinder natural frequency" competition (page #15) reminded me of the study of Bourguet and Triantafyllou (2016), that is also a very relevant work.

Thank you very much for pointing this out.

*****References*****
Balasubramanian, S., Haan, F. L., Szewczyk, A. A., & Skop, R. A. (1998).
On the existence of a critical shear parameter for cellular vortex shedding from cylinders in nonuniform flow. Journal of Fluids and Structures, 12(1), 3–15.
https://doi.org/10.1006/jfls.1997.0122
Balasubramanian, S., Haan, F. L., Szewczyk, A. A., & Skop, R. A. (2001). An experimental investigation of the vortex-excited vibrations of pivoted tapered circular cylinders in uniform and shear flow. Journal of Wind Engineering and Industrial Aerodynamics, 89(9), 757–784. https://doi.org/10.1016/S0167-6105(00)00093-3
Hover, F. S., Techet, A. H., & Triantafyllou, M. S. (1998). Forces on oscillating uniform and tapered cylinders in crossflow. Journal of Fluid Mechanics, 363, 97–114.

https://doi.org/10.1017/S0022112098001074

Bourguet, R., Modarres-Sadeghi, Y., Karniadakis, G. E., & Triantafyllou, M. S. (2011). Wake-body resonance of long flexible structures is dominated by counterclockwise orbits. Physical Review Letters, 107(13), 1–4. https://doi.org/10.1103/PhysRevLett.107.134502

Bourguet, R., Karniadakis, G. E., & Triantafyllou, M. S. (2013). Distributed lock-in drives broadband vortex-induced vibrations of a long flexible cylinder in shear flow. Journal of Fluid Mechanics, 717, 361–375. https://doi.org/10.1017/jfm.2012.576

Bourguet, R., & Triantafyllou, M. (2016). The onset of vortex-induced vibrations of a flexible cylinder at large inclination angle. Journal of Fluid Mechanics, 809, 111-134. doi:10.1017/jfm.2016.657

**Response to Reviewer 2**

The authors thank the Reviewer for their detailed and positive feedback. We have taken the feedback into account in the revised manuscript. Changes are highlighted in red. Each comment is further addressed below.

1) Introduction, Page 1: Provide some numbers as to what you consider as a large diameter or tall wind turbine for which VIV is relevant.
We have added some numbers in the introduction. We believe that towers of height between 65 and 110m are the most susceptible to VIV.

2) Section 2.1 - While you outline the k-w CFD model in detail, there is not much mentioned about the external wind conditions. What free wind condition range is applicable to your model in terms of free wind speed and turbulence? Can you also consider wind shear and ground effect?
Some more text is added regarding the limitations of the turbulence model. Within the two equations eddy viscosity model, there are only two parameters for the turbulence: one for the length-scale and another one for the intensity. There are some limitations for the boundary layer transition regime for the model. Also, the model is 2D so it does not take (vertical) wind shear into account.

3) Page 5: Why is there no aerodynamic damping term present in Eq. (18)?
We noticed that this equation should be expressed in terms of force coefficient with respect to y^star instead of y, so this has been corrected. The non-dimensional force coefficient implicitly contains the aerodynamic damping.

4) Page 7: From figure 2, it appears that bending in only one direction is considered. For a wind turbine tower, both side-side and fore-aft modes are excited in VIV and so at least two springs in perpendicular directions should be considered.
This is correct. However, this study is limited to VIV in one direction. This is now made clearer both in the text and the abstract. Although this is a limitation in the context of wind turbines, our analysis helps understanding the system dynamics in the transverse direction and also does bring new insights into VIV at flow conditions that are encountered in wind energy.

5) Section 3:0 : Can you provide a figure of the CFD mesh you used?
The mesh has a "standard" o-grid topology which is commonly used for flow around cylinders. Two figures are added to the text to illustrate the mesh.

6) Page 8: You state the structural damping is 0.007. How much artificial damping due to the CFD mesh do you generate in your model? Is this artificial damping significant with respect to your structural damping?
We did a mesh uncertainty study and the chosen mesh should not have a significant influence to the results as stated in the manuscript.

7) Same question for the turbulent flow: How much artificial/numerical damping is present in your CFD model and what affect does that have on the results?

This is a very good question but a tricky one to answer. We expect the mesh to have the highest influence on the results. As stated above, we have performed a mesh convergence study in order to limit the influence of the mesh on the results. Of course, other uncertainties are also present due e.g. to the level of convergence, time scheme, round of error, etc. These are expected to be smaller. The fluid-structure interactions of course also add uncertainties. However, the strong coupling scheme ensures that both fluid and structural solvers are in equilibrium at each time step.

8) Going by the results of Table 2 on the angle of flow separation over the cylinder, what would be the best direction to orient the spring for your structural oscillation, since you consider only single dimensional oscillations?

The pure vortex induced oscillations are investigated in this research. Coupling of both modes or other phenomena are not of interest as the resultant deflections (bending moment resulting from cross wind VIV) are the highest.

9) Page 12, line 240: The expression for aerodynamic damping used is not clear. There is no 'q' term in the equation as given in the explanation.

'q' was referring to a generic quantity, in order to explain what the prime and overbar denote. Here q is y_dot. This is now made clearer in the text.

10) Figure 7, Figure 8 etc: Can you also plot this versus the Strouhal number?

The Strouhal number is now used on the x-axis of the figures. This is indeed also how it was done in the NASA report used as reference. The values on the plot are unchanged.

11) Section 3.2.4: When you state realistic wind turbine tower, what is the wind turbine tower diameter, height and natural frequency that is considered?

We work with non-dimensional numbers throughout the paper. These are computed based on the properties of a real wind turbine tower from Siemens Gamesa Renewable Energy. The industrial partner prefers not to mention the dimensional values. However, the manuscript is scientifically complete and reproducible with the non-dimensional values.

12) Can you conclude on how the results of your work can be applied to an existing wind turbine tower? What wind conditions and tower natural frequencies should the turbine designer pay attention to for VIV?

We have added some text related to this in the conclusion. The current research gives us a great insight into the flow behavior and the perseverance of VIV in the supercritical Reynolds number when the flow regime is very stable. It explains episodes of sustained VIV where the Strouhal relation is not even valid. The magnification of VIV for a disturbed tower (oscillating from other wind phenomena such as gust buffeting or blade-tower interaction) under the right conditions can result in continuous magnifying VIV until the wake completely breaks down. The best mitigation strategy is not in the primary structure design but having robust damping mitigation

which are effective within the range of first bending frequency {0.5 – 1.1} to prevent the onset of sustained VIV.

[revised manuscript text omitted]

---

## Author Response (AR2)

The authors would like to thank the Editor for their feedback and comment. We hope that our revisions are satisfactory. Changes are highlighted in blue in the manuscript. Each comment is further addressed below.

Editor: Lines 27-30 – the materials and thicknesses also significantly affect the tower natural frequencies… so concrete tower and hybrids for example will behave differently. Thus, the conclusion that towers of 65 to 110 m are susceptible depend on additional factors… see for example https://www.nrel.gov/docs/fy18osti/70642.pdf (just an example since I have it on hand)

Response: The sentence is changed to specify that the statement concerns towers made of steel.

Editor: Line 31 - Tapered cylinders is correct but typically they have sections of discrete diameters (minor note, the above reference also models them as continuous tapered cylinders)

Response: This is added to the text.

Editor: Line 33 – the exclusion of taper is a key assumption… hopefully the limitations of this assumption are thoroughly discussed

Response: The effect of tapering certainly needs to be investigated further. We added in the text that this effect has already been investigated to some extent in the literature, although at smaller Reynolds numbers. Here, the focus of the paper is rather on high Reynolds numbers covering the supercritical regime, which is also of high relevance to the field, and for which publications are lacking even for circular cylinders. Note that the industrial partner co-authoring the study is also particularly interested in cylinders without tapering, because of their relevance for future tower designs. Thus, despite the limited scope of the study, the present outcomes are still highly relevant for the wind energy industry. We also mention the need to investigate tapering further in the conclusions.

Editor: 35 – several decades ago

Response: Corrected.

Editor: Figure 1 is quite small

Response: The size has been increased.

Editor: Figure 2 text is very small and hard to read

Response: The figure has been done again with bigger text size.

Editor: Section 4 Conclusions – the future work portion could be strengthened significantly. Also, the conclusions fail to tie the paper and results back to the context application of turbine VIV during installation. This could be improved considerably.

Response: The conclusion has been revised in order to expand on the future work portion and also provide more links with the application to wind turbine installations.

---

## Author Response (AR3)

Dear Editor in Chief,

We would like to thank again the Associate Editor. Her comment to put the paper in a better context is now clearer to us. We have therefore amended both the introduction and conclusion of the paper to discuss certain tower design aspects in more details. Changes are highlighted in green and are summarized hereafter.

On page 2, we remove the reference to steel towers and emphasize that the orders of magnitude given (which were asked by another Reviewer) are very much property-dependent. In this respect, we reference the paper suggested by the Associate Editor. Additionally, we highlight the fact that the present study is also motivated by the current trend to decrease the frequency of the first tower mode of vibration.

On page 20, we add some current trends in the design of wind turbine towers, as discussed in the paper recommended by the Associate  Editor. We further highlight the challenges in terms of frequency decrease and relevance for the installation phase of wind turbines.

We hope that these changes put our study a little bit more in the context of wind turbine tower design and we thank again the Editor for her advice to improve the quality of our manuscript.

Best regards,
Axelle Viré